# Delayed entanglement echo for individual control of a large number of nuclear spins

Zhen-Yu Wang[1], Jorge Casanova[1] & Martin B. Plenio[1]

Methods to selectively detect and manipulate nuclear spins by single electrons of solid-state defects play a central role for quantum information processing and nanoscale nuclear magnetic resonance (NMR). However, with standard techniques, no more than eight nuclear spins have been resolved by a single defect centre. Here we develop a method that improves significantly the ability to detect, address and manipulate nuclear spins unambiguously and individually in a broad frequency band by using a nitrogen-vacancy (NV) centre as model system. On the basis of delayed entanglement control, a technique combining microwave and radio frequency fields, our method allows to selectively perform robust high-fidelity entangling gates between hardly resolved nuclear spins and the NV electron. Long-lived qubit memories can be naturally incorporated to our method for improved performance. The application of our ideas will increase the number of useful register qubits accessible to a defect centre and improve the signal of nanoscale NMR.

[1] Institut für Theoretische Physik and IQST, Universität Ulm, Albert-Einstein-Allee 11, D-89069 Ulm, Germany. Correspondence and requests for materials should be addressed to Z.-Y.W. (email: zhenyu3cn@gmail.com) or to J.C. (email: jcasanovamar@gmail.com) or to M.B.P. (email: martin.plenio@uni-ulm.de).

Nuclear spins are natural quantum bits with long coherence times for quantum information tasks[1] and they encode information about the structure of molecules and materials in a form that is accessible to nuclear magnetic resonance (NMR) techniques[2] or electron-nuclear double resonance (ENDOR)[3]. However, because of the weak magnetic moments of nuclear spins and their similar precession frequencies, it is challenging to detect and control nuclear spins individually. The nitrogen-vacancy (NV) centre in diamond represents a promising nanoscale platform for detection and coherent control of such nuclear spins[4–6]. In type IIa diamonds, the decoherence of the NV electron spin is dominated by the presence of $^{13}$C nuclei. However, when properly controlled, the $^{13}$C nuclei in the vicinity of an NV centre become useful resources[7–9] for quantum computing purposes. Furthermore, shallow implanted NV centres can be used to detect the signal of nuclear spins on the surface[10,11], which opens opportunities for both quantum simulation[12] and single molecule NMR[11].

Using magnetic resonance techniques originally developed in NMR, nuclear spins have successfully been detected by single NV centres[6,10,11,13–19]. Among them, dynamical decoupling (DD) techniques[2,20] provide remarkable advantages, including significant extension of qubit coherence times and, more importantly, the ability of addressing single nuclear spins[13–15,21,22]. Nevertheless, standard DD techniques can only be used to address a few nuclear spins because of a number of drawbacks such as low spectral resolution[23], resonance ambiguities[24,25] and perturbations from the electron-nuclear coupling[21]. In this respect ENDOR techniques are applied to improve the spectral resolution by measuring the NV signal over long evolution times[23,26,27], but without the abilities of individual spin addressing and control. The main reason of the lack of selective addressing comes from the unfiltered coupling to undesired nuclear spins such as those in the spin bath. Moreover, in spin detection the presence of undesired coupling unavoidably produces noise on the sensor electron and reduces the sensitivity. For example, the signals from the target nuclear spins which has weaker coupling can be damaged by those with stronger coupling, regardless of the potential high spectral resolution. As a consequence reported results using the ENDOR techniques[18,23,26,27] provide high spectral resolution but do not demonstrate any increase in the number of detectable nuclear spins by sensor electrons comparing with the achieved numbers by standard DD techniques[6,14]. Note that individual addressing of nuclear spins and the implementation of quantum gates on them are more demanding than nuclear-spin detection, but essential in quantum information processing and thorough characterization of nuclear spins.

Our method overcomes these difficulties to selectively address specific nuclear spins by applying radio frequency (RF) fields in a delay window, while the entanglement with the electron spin sensor is preserved by a subsequent Hahn echo operation[2]. In this manner highly selective entangling gates between the electron spin and different target nuclear spins can be achieved via selective double resonance. We demonstrate that with our method one can address, control, and identify nuclear spins and spin clusters that are not detectable by standard DD techniques. Additionally, our protocol provides the ability to identify the number of nuclear spins sharing an unresolvable signal dip in the resonance spectrum. Furthermore, we show how one can individually address nuclear spins to perform high-fidelity two-qubit quantum gates between the NV electron qubit and the addressed nuclear qubit in a robust way, even when they are hardly resolved in spectrum. We also show how the efficiency of our method can be enhanced by using a nuclear memory, which has coherence times much larger than the life time of the electron spin of the sensor.

## Results

**Working principle**. Now we describe the details of our method. A magnetic field $B_z\hat{z}$ parallel to the NV symmetry axis splits the spin triplet of the orbital ground electronic state of the NV centre. We use two of the three levels $m_s = 0, \pm 1$ to define an NV electron spin qubit[4]. Under a strong magnetic field such that the nuclear Zeeman energies exceed the perpendicular components, $A_j^\perp$, of the hyperfine field $\mathbf{A}_j$ at the locations of nuclear spins, see Fig. 1a, the interaction between the NV electron spin and its surrounding nuclear spins is described by (Supplementary Note 1)

$$H_{int} = \sigma_z \otimes \eta \sum_j A_j^{\parallel} I_j^z \qquad (1)$$

with $\sigma_z = |\uparrow_e\rangle\langle\uparrow_e| - |\downarrow_e\rangle\langle\downarrow_e|$. Here we use $|\uparrow_e\rangle = |m_s = +1\rangle$ as one of the qubit states while the second qubit state may be $|\downarrow_e\rangle = |m_s = 0\rangle$, when $\eta = 1/2$, or $|\downarrow_e\rangle = |-1\rangle$ with $\eta = 1$. For each nuclear spin, $A_j^{\parallel}$ denotes the component of $\mathbf{A}_j$ parallel to the nuclear spin quantization axes (see Fig. 1a,b). The nuclear precession frequencies are shifted by $A_j^{\parallel}$, which can be used to address individually nuclear spins of the same species (homonuclear spins).

An initial superposition state of the electron spin $|\psi_e\rangle = c_\uparrow|\uparrow_e\rangle + c_\downarrow|\downarrow_e\rangle$ loses its coherence because of the electron-nuclear coupling $H_{int}$. This effect can be removed by a Hahn echo. In our case, a microwave $\pi$ pulse exchanges the states $|\uparrow_e\rangle \leftrightarrow |\downarrow_e\rangle$ and effectively reverses $H_{int} \rightarrow -H_{int}$. When the evolutions before and after the $\pi$ pulse have the same duration $\tau$, entanglement from all the nuclear spins is erased, preserving the electron spin coherence. In this manner, the nuclei are effectively decoupled from the electron. However, no nuclear spins can be addressed in this way.

To preserve the coupling with a target spin while eliminating the influence of the rest on our sensor, we apply RF driving at the precession frequency of the target spin during the delay window. By flipping the target spin $j$ by an angle $\theta_{rf} = \pi$, we selectively rephase its interaction with the electron spin and realize a quantum gate

$$U_{ent}(\phi_g) = |\uparrow_e\rangle\langle\uparrow_e| \otimes U_+ + |\downarrow_e\rangle\langle\downarrow_e| \otimes U_- \qquad (2)$$

with $U_\pm = \exp(\mp i\frac{1}{2}\phi_g I_j^z)$ and $\phi_g = 4\eta\tau A_j^{\parallel}$, up to single-spin rotations. Note that, by choosing a proper rotating frame, this is equivalent to write $U_+ = I$ and $U_- = \exp(i\phi_g I_j^z)$. Hence, we can generate entanglement between the NV centre and selected nuclear spins which forms a key ingredient of our method (see Fig. 1c,d). Note that it is not essential that $\theta_{rf} = \pi$ as almost any choice of $\theta_{rf} \neq 2m\pi$ ($m$ being an integer number) will create entanglement with the target and lead to a coherence dip in the electron spin spectrum. In addition, more than one nuclear spin can be addressed in the same delay window by using RF driving with different frequencies. Now, we describe two strategies to achieve selective nuclear spin control in the delay window.

**Spin addressing and high-fidelity quantum gate**. The first procedure corresponds to use DD techniques to suppress electron-nuclear interactions and to protect the NV coherence during the delay window, while the RF driving is applied to achieve selective rotations on the target nuclei (Fig. 1e). In nuclear spin sensing protocols, the electron qubit is initialized in an equally weighted superposition state $|\psi_e\rangle$ and its population signal at the end of the protocol $P = (1 + L)/2$ is directly related to the observable of NV electron coherence[15] $L = |\langle\downarrow_e|\rho_e|\uparrow_e\rangle|$ with $\rho_e$

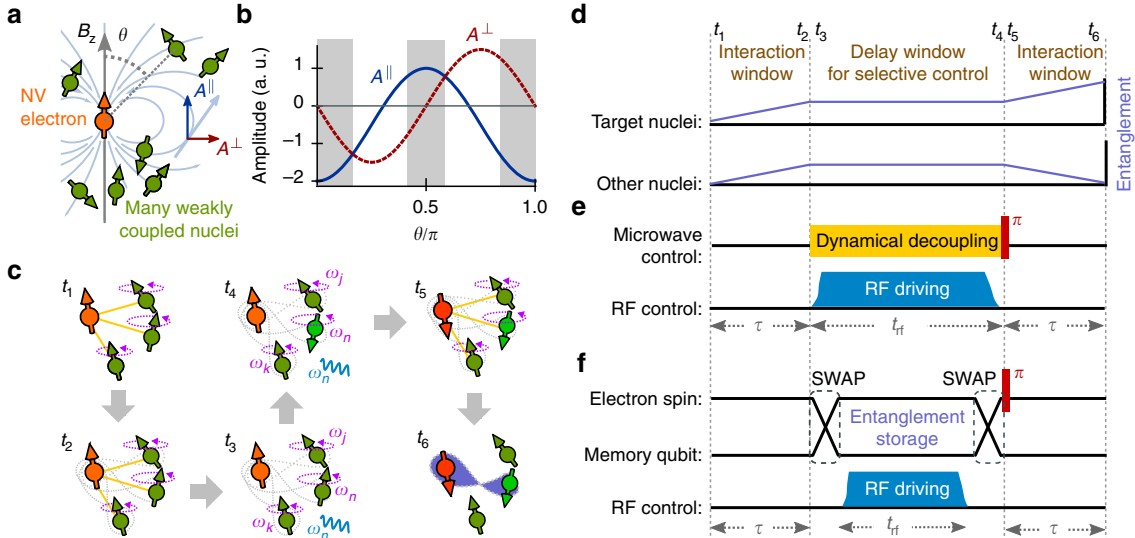

**Figure 1 | Delayed entanglement control between the electron and its surrounding nuclear spins.** (**a**) A large number of nuclear spins weakly coupled to the electron spin of NV centre by the hyperfine field. (**b**) Relative amplitudes of the components of the hyperfine field. The perpendicular component $A^\perp$ is weaker than the parallel component $A^\parallel$ at the shaded regions of the plot. (**c**) Illustration of the spin states in the process of delayed entanglement echo, with the entanglement changes sketched in **d**. Implementation of the delay window by DD (**e**) or by storing the entanglement to a long-lived memory qubit (**f**). During the delay window, the driving at RF frequencies provides highly selective control on nuclear spins for addressing. A $\pi$ pulse applied at the end of the delay window eliminates the coupling from unwanted nuclei as well as static noise.

being the reduced density matrix of electron qubit. Figure 2a,b shows the obtained spectrum when scanning the frequency of the RF driving applied at the delay window, for a diamond sample of natural abundance (1.1%) with 736 $^{13}$C spins. The NV coherence at the delay window is protected by the Carr–Purcell (CP) sequence[2]. Note that we can also use modified versions[21,28,29] of CP sequence by changing the phases of pulses to provide improved robustness at the present of control errors. When the RF frequency $\omega_{rf}$ matches one of the nuclear precession frequencies that, under strong magnetic fields, read $\omega_j = \omega_{^{13}C} - A_j^\parallel/2$ (with $\omega_{^{13}C}$ denoting the bare Larmor frequency of $^{13}$C) we observe the coherence contributed by the corresponding spin under on-resonant RF driving (see Methods)

$$L\left(A_j^\parallel, \theta_{rf}\right) = \sin^2\left(\frac{\theta_{rf}}{2}\right)\cos\left(2\eta A_j^\parallel \tau\right) + \cos^2\left(\frac{\theta_{rf}}{2}\right) \quad (3)$$

which lies in the range $\cos\theta_{rf} \le L(A_j^\parallel, \theta_{rf}) \le 1$. For the case of $\theta_{rf} = \pi$ we have a maximum signal $L(A_j^\parallel, \theta_{rf}) = \cos(2\eta A_j^\parallel \tau)$. When there are a number $p$ of nuclear spins with indistinguishable precession frequencies $\omega_j \approx \omega_{rf}$ the coherence signal becomes $L = [L(A_j^\parallel, \theta_{rf})]^p$. The one-to-one correspondence between $L$ and $\omega_j$ (see the brown dashed lines on Fig. 2 for $p = 1$ and the cyan dashed lines for $p = 2$, and compare the brown dashed lines with the signals of C1 and C3, as well as the cyan dashed lines with the dip of C2) and the coherence patterns in Fig. 2c,d easily identify the number of nuclear spins in a dip, even when they are not resolved in the spectrum. Because of the relation between the interaction strength and the spectral position of a signal dip, we know the electron-nuclear interactions when we set the RF driving at the frequency of the signal dip. In this manner, we can selectively control the nuclear spins creating the signal dip.

Our method works well even when the perpendicular components of hyperfine field $A_j^\perp = 0$ (see Fig. 1a,b). Note that standard methods[13–15,21,22,30,31] on spin detection based on DD techniques require both $A_j^\perp$ and $A_j^\parallel$ to be different from zero. In this respect, Fig. 2a shows the signal obtained with standard

DD[13–15] by scanning the pulse interval $\tau_{CP} = \pi/\omega_{DD}$ of the CP sequences. The coherence signal $L \approx \cos[A_j^\perp t_s/(k_{DD}\pi)]$ is determined by the strength of $A_j^\perp$ for a sequence duration $t_s$, when the $k_{DD}$-th harmonic branch matches the precession frequency $\omega_j$ (that is, $k_{DD}\omega_{DD} = \omega_j$)[22]. Therefore, nuclear spins with $A_j^\perp = 0$ (for example, C2 and C3 in Fig. 2a) are not detected by standard DD techniques. Furthermore, in Fig. 2a the signal obtained by DD is weak when compared with our method for the same number of 100 $\pi$ pulses. See Supplementary Note 2 for a discussion of standard DD methods, a more detailed comparison with our technique, and other shortcomings of standard DD techniques.

In addition, our protocol allows to enhance the interaction and, therefore, the obtained signal by transferring the electronic state $|\downarrow_e\rangle = |0\rangle$ to $|\downarrow_e\rangle = |-1\rangle$ for the interaction windows $(t_1, t_2)$ and $(t_5, t_6)$ in Fig. 1, with the results shown in Fig. 2b. In contrast, homonuclear spins can not be distinguished by the standard DD methods[13–15] when the qubit states $m_s = \pm 1$ are used, because the averaged nuclear precession frequencies are not shifted by the hyperfine field (Supplementary Note 2) and are not resolvable in spectrum.

Figure 2b also shows that a change in the rotation angle $\theta_{rf}$ due to, for example, intensity fluctuations in the RF driving field does not affect the locations of signal dips, demonstrating an intrinsic robustness of our method for nuclear spin detection. The fringes that appear around $\theta_{rf} = 2\pi$ are caused by the side excitation of the rectangular RF driving with a finite length $t_{rf}$. The RF driving with a frequency detuning $\Delta_{rf,j} \equiv \omega_{rf} - \omega_j$ turns the nuclear spin around an axis out of the $\hat{\mathbf{x}} - \hat{\mathbf{y}}$ plane with a tilting angle $\alpha_j \equiv \arctan(\Delta_{rf,j} t_{rf}/\theta_{rf})$ and an effective rotation angle $\tilde{\theta}_{rf,j} \equiv \sqrt{\theta_{rf}^2 + (\Delta_{rf,j} t_{rf})^2}$. For example, for $\theta_{rf} = 2\pi$ there is a spin flip ($\tilde{\theta}_{rf,j} = 3\pi$) around the tilting axis if the detuning is $\pm\sqrt{5}\pi/t_{rf}$, giving the side fringe signal shown in Fig. 2b. This single-spin contribution can be completely eliminated when $\tilde{\theta}_{rf,j}$ is an integer multiple of $2\pi$, even $\omega_j$ is not far detuned from the RF frequency. For example, for $\theta_{rf} = \pi$ a spin with a frequency detuning $\Delta_{rf,j} = \pm\sqrt{3}\pi/t_{rf}$ is rotated by the

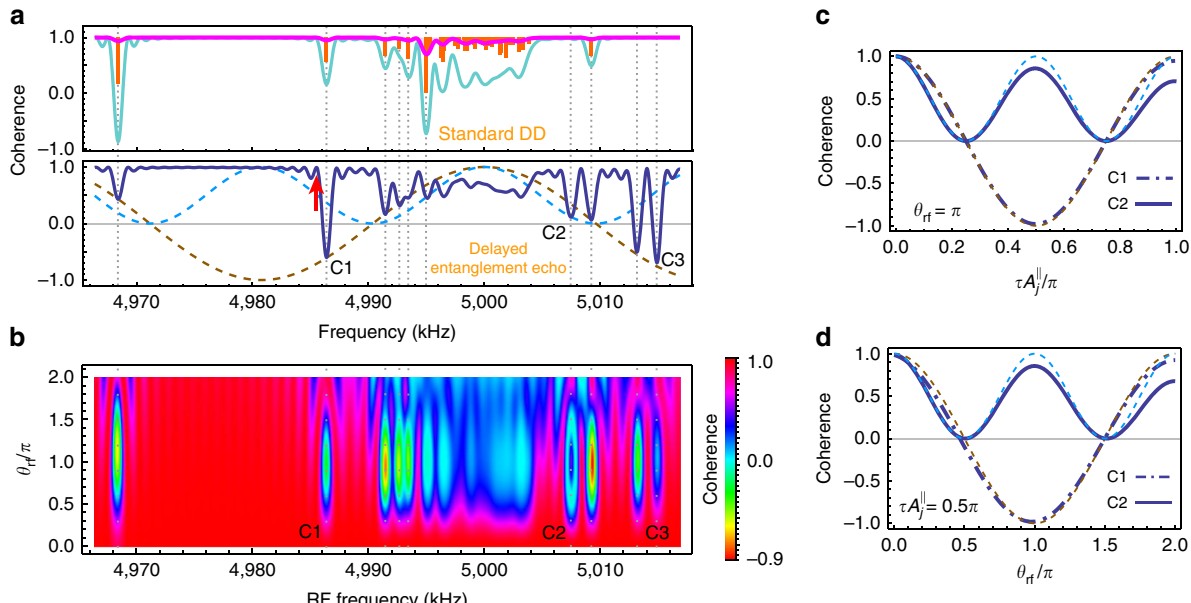

**Figure 2 | Coherence signals of delayed entanglement echo.** (**a**) Comparison between the delayed entanglement echo and standard DD method on the coherence dips of NV electron qubit by using the electron spin levels $m_s = 0$ and $+1$. The magenta (turquoise) solid line shows the population signal of standard DD method by 100-pulse (400-pulse) CP sequences as a function of the frequency $k_{DD}/(2\tau_{CP})$, where the harmonic branches $k_{DD} = 25$ (99) are chosen for a sequence duration of $\approx 1$ ms. The blue solid line shows the signal of delayed entanglement echo with the interaction time $\tau = 13$ μs, and the RF driving for the single-spin rotation angle $\theta_{rf} = \pi$ is protected by 100-pulse CP sequences to achieve a duration of $\approx 1$ ms. The dashed lines show the expected height of the signal dips $L^p(\omega_{rf} - \omega_{13_C}, \theta_{rf})$ for the delayed entanglement echo (see equation (3)) when there are one ($p = 1$, brown) or two ($p = 2$, cyan) spins at the corresponding spectral position. The lengths of the vertical orange lines are proportional to $A_j^\perp$, and the vertical dotted lines show the precession frequencies $\omega_j$ (bare Larmor frequency $\omega_{13_C} = 2\pi \times 5$ MHz). (**b**) Signal of delayed entanglement echo as in **a** but changing $\theta_{rf}$ and transferring the NV electron qubit to the levels $m_s = \pm 1$ for the interaction windows. (**c**,**d**) Coherence oscillations when changing $\tau$ and $\theta_{rf}$ using the scheme in **b**.

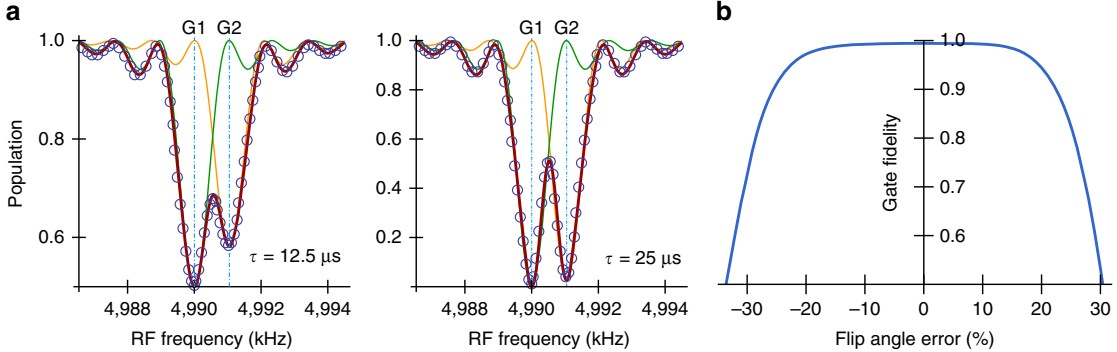

**Figure 3 | Single-spin addressing at the presence of control errors.** (**a**) Population signals (circles) of the delayed entanglement echo. Equally spaced AXY-8 sequences with a total number of 80 $\pi$ pulses (pulse length 50 ns and amplitude error 10%) are applied at the delay window, accompanied with the rectangular RF driving of the duration $t_{rf} = 0.827$ ms. Solid lines are the signals of G1 and/or G2 spins when the delay window is protected by ideal CP control. (**b**) Fidelity of combined quantum gate as a function of the flip angle error, with the G1 spin being addressed. The quantum gate on the G1 spin is an unconditional $\pi$ gate followed by a logical entanglement gate $U_{ent}(\phi_g)$ with the NV electron qubit. Here $\tau = 12.5$ μs to achieve $\phi_g = \pi$ for a maximum entanglement gate. The no operation (NULL) gates implemented on the G2 and the intrinsic $^{14}$N spins are included in the calculation of the combined gate fidelity. Without the interaction window ($\phi_g = 0$) the gate on G1 is an unconditional gate. In both **a**,**b**, the unpolarized $^{14}$N spin introduces a microwave detuning error of $\sim 2\pi \times 2.2$ MHz, but the combined fidelity achieves a value of 0.992 even for a flip angle error of 10%. The bare Larmor frequency $\omega_{13_C} = 2\pi \times 5$ MHz.

rectangular RF driving by an angle $2\pi$. Hence, the NV centre and the nuclear spin remain disentangled and a coherence dip is not produced (for example, see the red arrow near the main C1 coherence dip in Fig. 2a).

To investigate the effect of control errors from DD pulses used in our method, in Fig. 3 we consider a sample with two spectrally close $^{13}$C spins (G1 and G2). Figure 3a shows the population signals obtained by our method; during the delay window we

apply ideal instantaneous decoupling control on the NV centre in the form of CP sequences (red solid lines), while the blue circles correspond to the signal when errors in the external control are present. In the latter case we modify the phases of corresponding CP pulses to the ones of adaptive XY-8 (AXY-8) sequences[21] because AXY-8 sequences have good robustness against control errors. As shown in Fig. 3a, the case with errors fits well with the ideal signal even when we introduce a 10% of amplitude error on

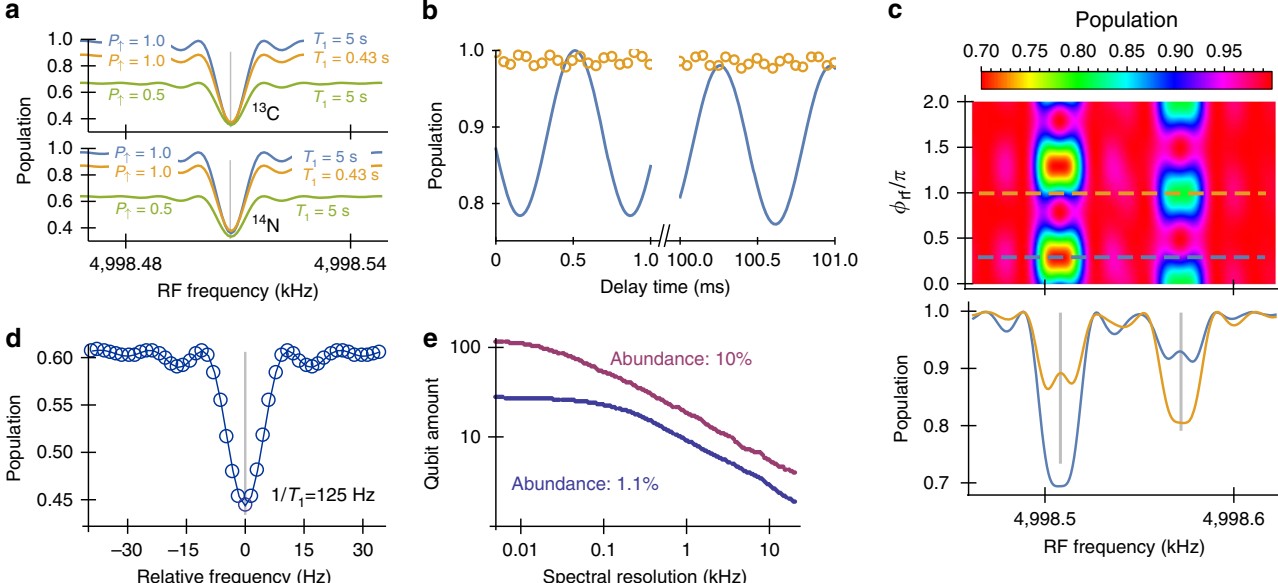

**Figure 4 | Delayed entanglement echo by using a nuclear memory qubit. (a)** Signal dips when addressing to a single distant $^{13}$C nuclear spin $\approx 2.2$ nm away from the NV centre by storing the qubit state in a $^{13}$C or in the intrinsic $^{14}$N memory qubit during the delay windows, for $t_{rf} = 100$ ms, $\tau = 100 \mu$s, and various low-temperature electron spin relaxation times $T_1$ and initial populations $P_\uparrow$ of the memory qubit state $|\uparrow\rangle$. **(b)** Population modulation (solid line) caused by two interacting $^{13}$C nuclei in a C–C bond $\sim 1.5$ nm away from the NV centre with $\tau = 200 \mu$s and a $^{14}$N memory. Application of LG decoupling suppresses nuclear dipolar coupling and hence the modulation (circles). **(c)** Signal patterns from two uncoupled distant $^{13}$C spins (2.2 and 2.3 nm away from the NV centre) when using a $^{14}$N memory qubit and DD to protect the interaction windows of duration $\tau = 500 \mu$s. The cross-sectional plots show the signals in the two-dimensional plot (dashed lines) when the initial RF phases $\phi_{rf}$ match the azimuthal angles of addressed nuclear spins. **(d)** Room-temperature signal of one proton spin placed 4 nm away from the NV centre using a $^{13}$C memory qubit and optical illumination during the delay window (Supplementary Note 5). With the RF driving of $t_{rf} \approx 10 T_1$, the line-width of the signal is well beyond the limit set by electron spin relaxation time $1/T_1$. One major reason of the reduction of signal contrast is the leakage out from the electron qubit to another electron spin triplet level. **(e)** Log-log plot of the average number of individual $^{13}$C register qubits with $A_j^{\parallel}/(2\pi) > 4$ kHz and $A_j^{\parallel}/(2\pi)$ different from other spins by amounts larger than the spectral resolution, by averaging over 1000 samples. A magnetic field is used for the bare Larmor frequency $\omega_{13_C} = 2\pi \times 5$ MHz. In **a,d,c**, $\theta_{rf} = \pi$, and $\theta_{rf} = 0$ in **b**. In **a,b**, the electron qubit state $|\downarrow_e\rangle = |0\rangle$ is transferred to $|\downarrow_e\rangle = |-1\rangle$ during the interaction windows.

the DD pulses, and a microwave detuning error of $\sim 2.2$ MHz caused by uncertain spin states of the strongly coupled $^{14}$N spin. As discussed above when we tune the RF-driving length to $t_{rf} = \sqrt{3}\pi/|\omega_{G1} - \omega_{G2}|$, the spin G2 does not perturb the dynamics of the NV electron and G1 spins. This is a situation that can be confirmed by changing the time $\tau$ of the interaction window and observing that the population change of the spin G1 follows the one of a single spin (see the pattens in Fig. 2c and compare the panels of Fig. 3a). With this value of $t_{rf}$, we can perform high-fidelity quantum gate on only one of the spins, even through the two spins G1 and G2 are not well separated in spectrum (see Fig. 3a). In Fig. 3b, we address the spin G1 by using the RF control $\omega_{rf} = \omega_{G1}$ and $\theta_{rf} = \pi$. The RF driving implements a $\pi$ rotation gate $R_{G1} = \exp(-i\pi I_{G1}^x)$ around the $\hat{x}$ direction, unconditional on the NV electron qubit state. If $\tau > 0$, the entanglement echo subsequently implement an entangling gate $U_{ent}(\phi_g)$ (see equation (2)) on G1. Figure 3b shows the fidelity (see Supplementary Note 5 for the definition) of the implemented maximally entangling gate $U_{ent}(\pi) R_{G1}$, achieving a combined gate fidelity of 0.992 even for a 10% amplitude and $\sim 2\pi \times 2.2$ MHz detuning errors on the $\pi$ pulses in the delay window.

**Enhanced performance via a quantum memory.** An alternative strategy to achieve selective nuclear spin control in the delay window is to store the electron spin state in a long-lived nuclear spin (see Fig. 1f). In Fig. 4a, we show the NV electron spin population signal when it addresses an isolated $^{13}$C spin, by using an initially polarized nuclear memory qubit. The memory qubit can be initialized by swapping a polarized NV electron state to the memory spin or by using dynamical nuclear polarization[16]. Because after a swap operation (Supplementary Note 3) between the NV electron and memory qubits the electron spin gets polarized to the $m_s = +1$ state during the delay window, it is not necessary to protect electron coherence. In addition, the shifts of the nuclear spin precession frequencies (now $\omega_j = \omega_{13_C} - A_j^{\parallel}$) under strong magnetic fields) are larger than those achieved by the method using DD or by standard DD techniques[13–15,21,22].

The electron spin relaxation creates magnetic noise on the nuclear spins and can reduce the signal contrast[32]. However, at low temperatures, the relaxation time $T_1$ of NV electron spin can reach minutes[33]. When $T_1 \gg t_{rf}$ the effects of electron spin relaxation can be neglected. In addition, we can protect the memory qubit against the electron-relaxation noise for example by strong driving[32]. The coherence times of memory qubits (for example, a $^{13}$C or a $^{14}$N in Fig. 4a) are determined by the electron-relaxation time $T_1$ when their hyperfine fields are sufficiently larger than $1/T_1$ (Supplementary Note 3)[32]. Therefore, the signal contrasts in Fig. 4a are similar for both memory qubits and we may use a more strongly coupled $^{13}$C spin as a quantum memory. Note that the use of unpolarized memory qubit still gives large signal contrast, showing that in the application of spin detection our method is insensitive to initialization error of the qubit memory (see Fig. 4a).

Structural information of coupled spin clusters can also be observed by our method. The dynamics of two coupled homo-nuclear spins under a strong magnetic field is characterized by the dipolar coupling strength $d_{j,k}$ (see Methods) and the difference

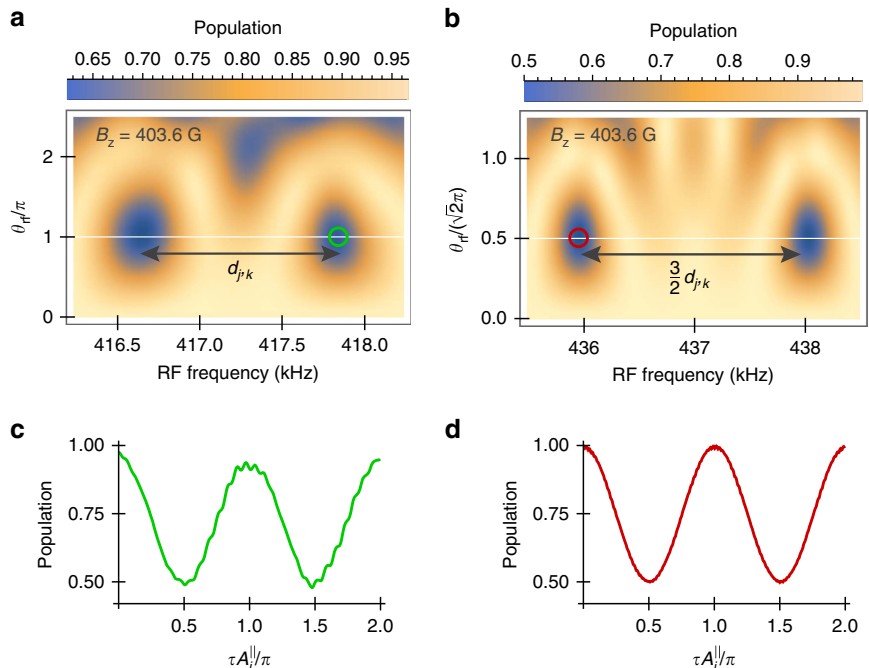

**Figure 5 | Population signal of coupled spin pairs. (a)** Signal patterns of a type-h $^{13}$C spin pair in a C–C bond $\approx 1.0$ nm away from the NV centre, using the interaction time $\tau = 10$ µs and length of RF driving $t_{rf} \approx 2$ ms. The dipolar coupling strength $d_{j,k} \approx 2\pi \times 1.37$ kHz. The hyperfine components of the two spins $A_j^{\parallel} \approx 2\pi \times 15.6$ kHz and $A_k^{\parallel} \approx 2\pi \times 22.5$ kHz. **(b)** Same as **a** but for a type-d pair located $\approx 1.55$ nm away from the NV centre and $\tau = 50$ µs, and the hyperfine components of the two spins $A_j^{\parallel} \approx -2\pi \times 5.00$ kHz and $A_k^{\parallel} \approx -2\pi \times 4.96$ kHz have similar strengths. The parameters used in **c,d** are indicated by the circles in **a,b**, respectively.

$\delta_{j,k} = A_j^{\parallel} - A_k^{\parallel}$ between the hyperfine components[34]. A spin pair close to the NV centre with $\delta_{j,k}$ and $d_{j,k}$ the same order of strengths can modulate the NV coherence[34,35], in the absence of RF control as shown in Fig. 4b. Spin pairs with $|\delta_{j,k}| \gg |d_{j,k}|$ (we call this as type-h) or $|\delta_{j,k}| \ll |d_{j,k}|$ (called type-d) could be useful quantum resources. But these two types of spin pair were regarded as unobservable, because they form stable pseudo-spins which have negligible effects on electron coherence[34]. However by using our method, we can detect, identify and control these spin pairs close to the NV centre. After the delayed entanglement echo with on-resonant RF driving, the population signal (see Methods) of a type-h or a type-d spin pair $P = 1/2 + [1 + \cos(2\eta A_j^{\parallel} \tau)]/4 \geq 1/2$ is different from the case of single spin, as shown in Fig. 5. The type-h and type-d spin pairs can be distinguished. For type-h pair the on-resonant RF field flips a $^{13}$C spin by $\theta_{rf} = \pi$, while for type-d pair the on-resonant RF field drive the triplet state with a Rabi frequency $\sqrt{2}$ times as the one for single nuclear spins (compare Fig. 5a,b and see Methods).

To address and control nuclear spins individually in a coupled cluster the internuclear dipolar coupling needs to be suppressed. Internuclear interactions also reduce the Hahn-echo electron coherence times and, hence, the achievable interaction times $\tau$ (available values[36,37] $\tau \sim 0.5$ ms for natural abundance of $^{13}$C and can be increased using lower abundance, see Supplementary Fig. 7). To solve the problem of spin interactions, we use the Lee–Goldburg (LG) off-resonance decoupling[2,12,22]. The LG decoupling field can remain turned on for the entire duration of our protocol, including NV electron spin initialization and readout, because the frequency of RF decoupling field is far off-resonance to the transition frequencies of the NV electron spin. This allows for the RF decoupling field to be applied by external coils and resonators to avoid possible heating on the diamond sample. When the LG decoupling field is tuned such that

$\sqrt{2}\Delta_{LG} \gg d_{j,k}$, the dipole-dipole interactions between nuclear spins are suppressed[12,22], giving rise to the effective Hamiltonian $H_{int} \approx \eta \sum_j A_j^{\parallel} \cos\gamma_j \sigma_z \tilde{I}_j^z / \sqrt{3}$ with $\tilde{I}_j^z$ the nuclear spin operators projected along effective rotating axes (see Supplementary Note 1). Figure 4b demonstrates the effect of LG decoupling with $\Delta_{LG} = 2\pi \times 20$ kHz, which can be achieved by a RF field with the amplitude much smaller than the values of $\sim 0.1$ T in the control fields reported in refs 38,39. The suppression of the internuclear interactions allows us to achieve longer interaction times $\tau$ and RF-driving lengths $t_{rf}$ for single-spin addressing. The improved spectral resolution $\sim 1/t_{rf}$ leads to an increase of the number of individually addressable spins (see Fig. 4e).

In the case that there are other significant decoherence sources that are acting on the NV electron spin, our method can be combined with DD to further protect the NV coherence. Applying a CP sequence (or its modified versions[21,28,29] for better control robustness) with an inter $\pi$-pulse interval $\pi/\omega_{DD}$ and the sequence duration $\tau$ at each of the interaction windows, noise with frequencies slower than $\omega_{DD}$ is suppressed, and at the same time the electron spin couples to nuclear spins through the interactions $2\eta/(k_{DD}\pi)A_j^{\perp}\sigma_z I_j^{\varphi_j}$ when the $k_{DD}$-th harmonic branch matches the precession frequency $\omega_j$ up to a frequency uncertainty of $\sim 1/\tau$ (refs 14,21,22). The nuclear spin operators $I_j^{\varphi_j} = I_j^x \cos\varphi_j + I_j^y \sin\varphi_j$ depend on the azimuthal angles $\varphi_j$ of nuclear spins relative to the magnetic field direction. Nuclear spins unresolvable by the CP sequences may nevertheless have different precession rates. To ensure the same effective Hamiltonian at the two interaction windows, we apply a two-pulse CP sequence on the nuclear spins during the delay window to remove this inhomogeneity. Then adding a weak RF drive during the delay window allows us to address the target spins with high spectral resolutions.

In addition, our scheme allows to measure the spin directions $\varphi_j$, as detailed in the following. The azimuthal directional angle

$\phi_{\mathrm{rf}}$ of RF control fields is controlled by the RF phase in the rotating frame of nuclear spin precession. When the RF driven rotation does not commute with the effective interaction Hamiltonian, it breaks the erasing process of delayed entanglement echo on the target spins, and thereby, we address the nuclear spins in a highly selective way. On the other hand, when the RF rotation axis is parallel to the azimuthal angle of a target nuclear spin ($\phi_{\mathrm{rf}} = \varphi_j$ or $\varphi_j + \pi$), the RF driving commutes with the interaction window and the electron-nuclear entanglement is removed after the echo, giving vanishing signal dips. By measuring the RF phases $\phi_{\mathrm{rf}}$ which cause vanishing signal dips, we obtain the relative locational directions of nuclear spins (see Fig. 4c). Note that we can combine LG decoupling with DD using recently proposed protocols[22,40].

Our method allows to improve the spectral resolution beyond the limit set by the room-temperature electron $T_1$ when we decouple the nuclear memory from the electron spin at the delay window. The decoupling technique that uses optical illumination has been demonstrated to prolong the room-temperature coherence time of nuclear spin memory over one second ($\sim 267$ times of $T_1$)[32], because the optically induced NV ionization decouples the nuclear spins from the NV centre by a mechanism related to the motional averaging effect in NMR[32,41]. In Fig. 4d, we simulate the application of optical illumination and RF driving during the delay window to detect a proton spin placed 4 nm away from the NV centre with the interaction time $\tau = 100\,\mu s$ (Supplementary Note 5). A delay time $t_{\mathrm{rf}} \approx 80\,\mathrm{ms}$ already provide enough frequency resolution to detect a chemical shift of $\sim 1\,\mathrm{p.p.m.}$ for the applied magnetic field $B_z \approx 0.467\,\mathrm{T}$. Using a higher magnetic field can further improve the chemical shift detection. We can apply LG decoupling when there are more target spins and internuclear interactions. In addition, we can use DD to protect the interaction windows from noise for extending the interaction time $\tau$. Electron spin coherence time of shallow NV centre has reported values of $\sim 1\,\mathrm{ms}$ using continuous DD (spin lock)[11].

## Discussion

In summary, we have proposed a method to address and control nuclear spins which were regarded as undetectable or unresolvable. High-fidelity quantum gates on these nuclear spins can be achieved by our method even when the external control has relative large errors and the nuclear spins present a dense resonance spectrum. Therefore, by our method more nuclear spins become useful and robust quantum resources. For example, with more reliable nuclear spins it is expected an improvement in quantum metrology assisted by quantum error correction[42]. The ability to address and control nuclear spins individually is also vital in the development of a solid-state quantum information processor that uses host electron spins and surrounding nuclear spins[40] as well as for large-scale quantum simulators in diamond[12]. Our method provides a substantial advancement over the ENDOR techniques[3,18,23,26,27] which do not individually address nuclear spins because of inefficient elimination of the noise from other bath spins (see Supplementary Note 4 for a discussion on the ENDOR sequences utilizing both microwave and RF control fields). As a consequence, our work is expected to find applications as well in traditional fields of NMR and ENDOR, for example, in analysis of chemical shifts. Finally, the method has a general character being equally applicable to other electron-nuclear spin systems[43,44].

## Methods

**Signals of delayed entanglement echo.** During the sensing protocol, the NV dynamics can be calculated by the methods in refs 37,45. Given the initial electron spin state $|\psi_e\rangle = (|\uparrow_e\rangle + |\downarrow_e\rangle)/\sqrt{2}$, the population left in the original equal

superposition state is $P = (1 + L)/2$ and the NV coherence $L = \prod_c L_c$ can be expressed by the contributions $L_c$ of spin clusters $c$ when the interactions between different spin clusters can be neglected[37,45]. Because $k_B/\hbar \approx 2\pi \times 21\,\mathrm{GHz\,K^{-1}}$, the thermal energies are much larger than the Zeeman energies of nuclear spins at relevant temperatures. Therefore the thermal state of nuclear spins is approximately described by the identity operator up to a normalization factor. The quantity $L_c$ reads

$$L_c = \frac{1}{2\mathcal{N}_c} \mathrm{Tr}\left( U_{c,+} U_{c,-}^\dagger + U_{c,-} U_{c,+}^\dagger \right), \tag{4}$$

where $U_{c,+} = \langle\uparrow_e|U_c|\uparrow_e\rangle$, $U_{c,-} = \langle\downarrow_e|U_c|\downarrow_e\rangle$, $\mathcal{N}_c$ the dimension of the spin cluster $c$, and $U_c$ the NV-cluster evolution[37,45].

**Single spins.** Single-spin dynamics dominate the effect on the NV electron spin[37]. The signal from single nuclear spins (a spin cluster $c = j$ with one spin $j$) is calculated as follows. The evolution of a single nuclear spin after an interaction window with time $\tau$ is $e^{-i\eta A_j^\parallel \tau \sigma_z I_j^z}$, where $\eta$ is determined by the qubit manifold used in the interaction window. During the delay window, we apply a RF driving field that rotates the nuclear spins with the evolution $R_j$. For resonant RF driving (that is, the RF frequency $\omega_{\mathrm{rf}} = \omega_j$ matching the nuclear precession frequency $\omega_j$), the nuclear spin is rotated by an angle $\theta_{\mathrm{rf}}$ and $R_j = e^{-iI_j^x \theta_{\mathrm{rf}}}$ (assuming rotation around the $\hat{\mathbf{x}}$ direction). For the rectangular RF-driving with a duration $t_{\mathrm{rf}}$ and a frequency detuning $\Delta_{\mathrm{rf},j} = \omega_{\mathrm{rf}} - \omega_j$, we obtain the spin rotation at the delay window $R_j = \exp[-it_{\mathrm{rf}}(\frac{\theta_{\mathrm{rf}}}{t_{\mathrm{rf}}} I_j^x + \Delta_{\mathrm{rf},j} I_j^z)] \equiv \exp[-i\bar{\theta}_{\mathrm{rf},j}(\cos\alpha_j I_j^x + \sin\alpha_j I_j^z)]$. Following by another interaction window with time $\tau$ between two $\pi$ pulses on the electron spin (the last $\pi$ pulse is optional, but we keep it for simplicity), we have the coupled evolution between the NV electron spin and a single nuclear spin $U_j = e^{i\eta A_j^\parallel \tau \sigma_z I_j^z} R_j e^{-i\eta A_j^\parallel \tau \sigma_z I_j^z}$, which gives the coherence signal by using equation (4). The contribution of a single spin-$\frac{1}{2}$ after the delayed entanglement echo with on-resonant RF driving is given by equation (3).

**Coupled spin pairs.** Strongly coupled nuclear spin pair (a spin cluster $c = \{j, k\}$ with two spins $j$ and $k$) can modulate the NV electron spin dynamics. Under a strong magnetic field the dipolar coupling reads $d_{j,k} = \frac{\mu_0}{4\pi} \frac{\gamma_j \gamma_k}{|\mathbf{r}_{j,k}|^3} [1 - 3(\hat{\mathbf{z}} \cdot \mathbf{r}_{j,k}/|\mathbf{r}_{j,k}|)^2]$, where $\mathbf{r}_{j,k}$ is the relative position of the coupled $j$ and $k$ spins and $\gamma_{j(k)}$ the gyromagnetic ratios. The coupled dynamics between the NV electron spin and the spin pairs of type-h ($|\delta_{j,k}| \gg |d_{j,k}|$) and type-d ($|d_{j,k}| \gg |\delta_{j,k}|$) under a strong magnetic field is shown as follows.

For type-h pairs, homonuclear spin flip processes are suppressed and therefore the precession frequency of each nuclear spin is split by $d_{j,k}$ by the internuclear interaction $H_{\mathrm{dip}} = d_{j,k} I_j^z I_k^z$. The nuclear spin flip by the RF driving at the delay window is conditional on the state of the other spin in the nuclear spin pair. Similar to the calculation of single spin, the evolution after the delayed entanglement echo is $U_{\{j,k\},\pm} = e^{-i\theta_{\mathrm{rf}} I_j^x \otimes |\uparrow\rangle\langle\uparrow|} e^{\mp i2\eta A_j^\parallel \tau I_j^z \otimes |\uparrow\rangle\langle\uparrow|}$, after an on-resonant RF driving $\theta_{\mathrm{rf}} = \pi$ to flip the nuclear spin $j$ conditioned on the state $|\uparrow\rangle$ of spin $k$. Different from the case of single spins, the dimension of the spin pair is $N_j N_k$, and the corresponding population signal for two $^{13}$C is $P = 1/2 + [1 + \cos(2\eta A_j^\parallel \tau)]/4 \geq 1/2$ when applying $\theta_{\mathrm{rf}} = \pi$ on resonant to the spin $j$.

For type-d pair of homonuclear spins, the interaction takes the form $H_{\mathrm{dip}} = d_{j,k}(3I_j^z I_k^z - \mathbf{I}_j \cdot \mathbf{I}_k)/2$ under a strong magnetic field. For the nuclear spins with spin $I = 1/2$, the composited spin cluster has a singlet state with a composited spin $J = 0$, $|s_n\rangle = (|\uparrow\downarrow\rangle - |\downarrow\uparrow\rangle)/\sqrt{2}$. The triplet states with $J = 1$ are $|1_n\rangle = |\uparrow\uparrow\rangle$, $|0_n\rangle = (|\uparrow\downarrow\rangle + |\downarrow\uparrow\rangle)/\sqrt{2}$, and $|-1_n\rangle = |\downarrow\downarrow\rangle$. A radio frequency control $H_{\mathrm{rf}} = \gamma_n B_x \cos(\omega_{\mathrm{rf}} t)(I_j^x + I_k^x)$ can be written as $H_{\mathrm{rf}} = \sqrt{2}\gamma_n B_x \cos(\omega_{\mathrm{rf}} t)(|1_n\rangle + |-1_n\rangle)\langle 0_n| + \mathrm{h.c.}$. Scanning the RF frequency $\omega_{\mathrm{rf}}$ to the splitting between $|0_n\rangle$ and $|\pm 1_n\rangle$, the nuclear spins are rotated. The nuclear state $|0_n\rangle$ has an energy of $-d_{j,k}/2$ (because $A_k^\parallel \approx A_j^\parallel = A^\parallel$), while the energies for $|\pm 1_n\rangle$ are $\pm\omega_n + d_{j,k}/4$, where $\omega_n$ is the nuclear precession frequency shifted by the hyperfine field. Therefore, the transition frequencies between $|0_n\rangle$ and $|\pm 1_n\rangle$ are $\omega_n \pm 3d_{j,k}/4$, shifted by the dipolar coupling. Different from the case of single spins that a spin flip requires a RF driving time $2\pi/(\gamma_n B_x)$, here the Rabi frequency is increased and transitions between $|0_n\rangle$ and $|\pm 1_n\rangle$ can be finished in a time of $\sqrt{2}\pi/(\gamma_n B_x)$. This time difference is a signature to distinguish the signals from that of single spins and of a type-h spin pair. Before a spin flip of the nuclear spin pair, the interaction Hamiltonian with the NV electronic spin reads $H = \eta\sigma_z(A_j^\parallel I_j^z + A_k^\parallel I_k^z)$, that is, $H = \eta\sigma_z A^\parallel(|+1_n\rangle\langle+1_n| - |-1_n\rangle\langle-1_n|)$. After a time $\tau$, we flip the electron spin and the nuclear triplet state, for example, with a transition $|0_n\rangle \leftrightarrow |+1_n\rangle$ in a delay window, giving an effective interaction $H = \eta\sigma_z A^\parallel(-|0_n\rangle\langle 0_n| + |-1_n\rangle\langle-1_n|)$ after the delay window for a time $\tau$. The join evolution up to single qubit operations reads $U = \exp[-i\eta\sigma_z A^\parallel \tau(|+1_n\rangle\langle+1_n| - |0_n\rangle\langle 0_n|)]$ and the corresponding population signal $P = 1/2 + [\cos(2\eta A^\parallel \tau) + 1]/4$.

**Data availability.** The data files used to prepare the figures shown in the manuscript are available from the first corresponding author upon request.

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

## Acknowledgements

This work was supported by the Alexander von Humboldt Foundation, the ERC Synergy grant BioQ, the EU projects DIADEMS, and EQUAM as well as the DFG via the SFB TRR/21. We thank Thomas Unden and Fedor Jelezko for discussions. Simulations were performed on the computational resource bwUniCluster funded by the Ministry of Science, Research and the Arts Baden-Württemberg and the Universities of the State of Baden-Württemberg, Germany, within the framework program bwHPC.

## Author contributions

Z.-Y.W., J.C. and M.B.P. conceived the idea. Z.-Y.W. carried out the simulations and analytical work with input from J.C. and M.B.P. All authors discussed extensively on the results and contributed to the manuscript.

## Additional information

**Competing financial interests:** The authors declare no competing financial interests.

**Publisher's note**: 

