## [Peer Review File · Nature Communications]

Reviewer #1 (Remarks to the Author):

The manuscript by Wang, Casanova, and Plenio investigated coherent control of remote nuclear spins around a single NV center using delayed entanglement echo technique. There are two key ideas in this work: (1) selectively addressing individual nuclear spins by combining microwave (mw) and radio-frequency (rf) fields and (2) further improving nuclear spin selectivity by storing the entanglement to a long-lived ancillary memory qubit. These ideas that can further improve the selectivity and controllability of more nuclear spins. Here are some comments and questions about this work:

1. The manuscript focuses on how to spectroscopically resolve many nuclear spins, which is a necessary step towards the claim of "individual control" of many nuclear spins. However, demonstrating spectroscopy resolvability does not necessarily imply controllability.
 - a. For example, in lines 158-161, the authors claim that using unpolarised memory qubit can improve the resolvability, but one still cannot claim controllability of the nuclear spin with unpolarised memory qubit. It will be helpful to clarify this statement and avoid potential confusion.
 - b. Moreover, in order to claim controllability of individual nuclear spins, it will be important to show a simulation of individual nuclear spin rotation with high-selectivity and high-fidelity (e.g., two nuclear spins, separated by 1 kHz, with selective rotation of one but not the other, under realistic parameters).

2. In lines 229-231, the authors claimed that they have simulated the combined optically illumination and rf driving during the delay window. Generally, optical illumination can induce very complicated transitions of the NV center (including ionization) that may affect the nuclear spin evolution, which leads to the following questions:
 - a. Which model is used for the simulation of the optical illumination?
 - b. How strong is the optical illumination?
 - c. Will optically induced NV ionization affect the nuclear spin coherence in the presence of rf driving?It will be important to include these details in the supplemental material in order to make the claim that combined optical illumination and rf driving can be useful.

3. In Fig.2, is there a particular reason why panel (a) and panel (b) choose different interaction times 10 us and 13 us?

Overall, I think that the paper presents interesting ideas that may further extend our capability of individually control more nuclear spins around a single NV center. If the authors can address the above comments and questions, I would like to recommend the publication of this paper in Nature Communications.

Reviewer #2 (Remarks to the Author):

The paper of Wang et al., proposes a method for the detection of nuclear spins which are weakly coupled to an electron spin - specifically that of a single NV center in diamond. The idea is based on using a long (and thus highly frequency-selective) RF pulse to drive the nuclear spins, applied either while an electron spin coherence is protected by dynamical decoupling, or while it is being stored in a nuclear spin memory.

A find two significant flaws in the current work which prohibit me to recommend publication.

- 1) This proposal falls within a methodology known as "Electron-nuclear double resonance", or ENDOR, in the field of magnetic resonance. A good reference for this is Chapter 12 of "Principles of

Pulse EPR" by Schweiger and Jeschke. The manuscript surprisingly does not make any reference to established ENDOR methods, including those which have been designed to detect nuclear spins which are weakly coupled to the electron spin. For example, at first inspection, I would expect the Mims ENDOR sequence (section 12.2.2 in the book cited above) to yield a similar result to that shown in Fig 2A. Furthermore, the proposed sequence bears a striking resemblance to "Hyperfine-decoupled ENDOR" (section 12.3.2), albeit using DD rather than spin-locking. I recommend that the authors perform a thorough review of established ENDOR methods (which go back 50 years), and then benchmark their proposal against these methods to determine what tangible advantages their approach might offer. As far as I'm aware, the element of their proposal which uses a nuclear spin memory may be novel, but the range of circumstances where that approach may be practically useful could be limited.

2) My second concern is that it would be typical in magnetic resonance, when a pulse sequence is proposed, to accompany that proposal with an experimental demonstration. Sometimes sequences which appear to work well on paper / in silico do not always perform as well in practical due to technical considerations. Therefore, it would be important, in my view, for a work such as this to include some experimental demonstration of the proposed technique, in order for it to attract a wide interest

Reviewer #3 (Remarks to the Author):

A. The manuscript describes a scheme enhancing the capability of addressing and controlling specific nuclear spins in solids, through their interaction with a probe electronic spin. The method is described in the context of an NV center in diamond coupled to other nuclear spins, mostly ^{13}C spins in the diamond lattice.

While such addressing and control has been demonstrated in the past (as mentioned, e.g., in refs. 3,4,22), the proposed scheme incorporates an additional RF driving of the nuclear spins to provide another control element for enhanced addressability and control.

The authors demonstrate theoretically the advantages of the proposed scheme, along with several improvements, such as combining with dynamical decoupling (DD) of the NV spin and using nearby nuclear spins as memory for better spectral resolution.

B. The proposed scheme could potentially improve significantly the ability to address and control numerous nuclear spin through the control of a single electronic spin, thus impacting applications ranging from quantum information processing to nuclear magnetic resonance. It is therefore of interest to a relatively large community.

While the precise features of the scheme are novel, it is strongly based on known approaches, essentially combining driving of the control spin (the electronic spin in this case) and the target spins (the nuclear spins in this case). Such approaches include DEER and ENDOR schemes from the field of NMR, which have been demonstrated with NV centers as well (e.g., PRL 113, 197601 (2014) and Science 339, 557 (2013)). Nevertheless, I believe that the combined techniques presented in the current paper fulfill the originality requirement.

C., D., E., G. The proposal is clearly presented, along with the numerical analysis and results (with a few caveats mentioned below). Except for the few issues mentioned below, the approach is sound, well written, and convincing. The references are rather comprehensive.

A few issues:

- The authors include DD pulses to enhance the coherence properties of the NV, as described e.g. on line 116. This is a common approach used extensively in the NV community and beyond. For nuclear spin sensing it is common to apply symmetrized sequences, such as the XY family, instead of the CP sequence, which usually results in spurious artifacts in the signal. Why do the authors

choose to simulate the CP sequence? They should consider pulse errors, which are usually a major concern in experimental realizations.

- In standard NV-based nuclear spin measurement schemes, direct RF driving of the nuclear spins is not used, but a signal can be extracted from the combined Larmor precession of the nuclei and their hyperfine interaction with the NV. The authors should clearly compare that approach with the proposed driven scheme, and present the signals on the same footing, e.g. in Fig. 2a.

- Fig. 2 is rather confusing and poorly explained. It is not clear to me what the different dashed lines in 2a present, and the triangular arrows in the C1-C3 labeling. How many nuclear spins were included in this simulation? In the caption it is mentioned that the parameters used in Fig. 2a are different than the ones in 2b. Why? It would have been better to use the same parameters.

Also, it's not clear to me why in Fig. 2b there's a distinct difference between 0 phase and 2π phase (in the vertical axis). The authors should explain this difference.

- In Fig. 3a the upper and lower panels describe different memory nuclear spins (^{13}C vs. N), but the results seem almost indistinguishable. This is surprising, and if not a mistake, should be explained in more detail.

- I felt that the details in the main text are lacking, with more detail present in the supplementary. The authors and editors should consider moving some detail from the supplementary to the main text.

- In the supplementary, line 50, it is stated that for ± 1 states the hyperfine shift of the Larmor frequencies vanishes. This is not trivial to me, and should be explained in detail, as it is an important aspect of the theory.

In summary, I believe that the paper presents an interesting and potentially important scheme for enhanced addressing and control of nuclear spins using an control electronic spin.

I would recommend publication of the manuscript if the comments above are fully resolved.

Summary of changes:

We have revised the manuscript along all the remarks from the Reviewers. To assist them, we include a pdf version where the introduced changes are highlighted in blue. Furthermore, and as Reviewer #3 asked, we have moved some materials from the original Supplementary Information to the main text, including the section on signals of delayed entanglement echo (now at Methods section) and the details on combining the interaction windows with dynamical decoupling (now at Results section). Additionally, technical details in the methods section of the original version, have been moved to the results section of the resubmitted version. We have added a new figure, Fig. 3, on the effects of control errors and quantum gates. Figure 2 has been revised to fit the suggestions from the Reviewers. Besides, in Supplementary Information, we have added a new subsection (II B) on the dynamics under CP control, a new subsection (III C) on the coherence time of the nuclear memory, and a new section (IV) on electron nuclear double resonance. We have also added more references. Finally, the presentation of the manuscript (including the abstract, introduction part, and the discussion section) has been revised to fit the style and requirements of Nature Communications.

Reply to reviewers:

Note: Reviewers' remarks are in blue colour.

Response to Reviewer #1:

We thank Reviewer #1 for her/his careful reading and constructive and insightful remarks. In the following we address all the Reviewer's comments and questions.

(1) Concerning:

"The manuscript by Wang, Casanova, and Plenio investigated coherent control of remote nuclear spins around a single NV center using delayed entanglement echo technique. There are two key ideas in this work: (1) selectively addressing individual nuclear spins by combining microwave (mw) and radio-frequency (rf) fields and (2) further improving nuclear spin selectivity by storing the entanglement to a long-lived ancillary memory qubit. These ideas that can further improve the selectivity and controllability of more nuclear spins. Here are some comments and questions about this work:"

we thank the Reviewer for the evaluation of our work.

(2) With respect to the concern:

“1. The manuscript focuses on how to spectroscopically resolve many nuclear spins, which is a necessary step towards the claim of "individual control" of many nuclear spins. However, demonstrating spectroscopy resolvability does not necessarily imply controllability.”

We thank the Reviewer for bringing up this important point. We agree with the Referee that, in general, demonstrating spectroscopy resolvability does not necessarily imply controllability, because the NV centre electron spin could still couple to “background” spins. However, in our method the population/coherence signals provide signatures of effective electron-nuclear interactions. A signal value of 1 means that the electron spin is decoupled from all the surrounding nuclear spins. This signal changes when several nuclear spins start to interact with the electron spin. In this respect, our method allows us to know how many nuclear spins are creating a certain signal dip (e.g., the signals of C2 and C3 in Fig. 2). Therefore, we know if we are coherently controlling one spin via the action of a well-defined quantum gate, or if the signal is created by the combined action of several spins. Indeed, we have demonstrated that we are able to perform a complete set of single- and two-qubit gates between the electron spin and a specific nuclear spin in the spin bath [see Eqs. (S33) and (S34) in Supplementary Information for an example of the gates we have demonstrated, and question (4) in this reply].

To further clarify this issue, we have added the new lines 155-160 in the main text.

(3) Concerning:

“a. For example, in lines 158-161, the authors claim that using unpolarised memory qubit can improve the resolvability, but one still cannot claim controllability of the nuclear spin with unpolarised memory qubit. It will be helpful to clarify this statement and avoid potential confusion. “

we have changed the potentially confusing sentence:

“Even for unpolarised memory qubit, the signal contrast is still large, showing that our method is insensitive to initialisation error of the qubit memory (Fig. 3a)”

by (in lines 282-286):

“Note that the use of unpolarised memory qubit still gives large signal contrast, showing that in the application of spin detection our method is insensitive to initialisation error of the qubit memory (Fig. 4a).”

(4) Following the reviewer’s suggestion,

“b. Moreover, in order to claim controllability of individual nuclear spins, it will be important to show a simulation of individual nuclear spin rotation with high-selectivity and high-fidelity (e.g., two nuclear spins, separated by 1 kHz, with selective rotation of one but not the other, under

realistic parameters).”

we have added a simulation with two nuclear spins separated by 1 kHz under realistic external control conditions (see Fig. 3 and the related lines 213-249 added in the main text of resubmitted version, as well as Sec. V C, lines 310-326, in Supplementary Information) to show individual nuclear spin rotation with high-selectivity and high-fidelity. We would also like to note that in our method, it is easy to achieve single-qubit gates on the target spins by using the interaction time $\tau=0$ at the interaction windows.

(5) Concerning the question:

“2. In lines 229-231, the authors claimed that they have simulated the combined optically illumination and rf driving during the delay window. Generally, optical illumination can induce very complicated transitions of the NV center (including ionization) that may affect the nuclear spin evolution, which leads to the following questions:

a. Which model is used for the simulation of the optical illumination? ”

In our simulations we used the 11-level model explained in ref. 32 [Science 336, 1283 (2012)] (ref. 1 in the previous version). This 11-level model includes ionization and deionization of the NV centre and produces results that fit the experimental observations. In the resubmitted version, we have more explicitly stated the model we used for the simulation in Sec V G of Supplementary Information (lines 354-357) and comment on the main text in lines 391-394 and lines 397-398.

(6) Concerning the question:

“b. How strong is the optical illumination? ”

We used a strong optical illumination, 64 times the saturation intensity I_{sat} , to preserve the coherence of the nuclear memory for the delay time of 80 ms. Note that the optical illumination of 30 mW ($\sim 38 I_{\text{sat}}$) in ref. 32 [Science 336, 1283 (2012)] has already extended the coherence time of a similar nuclear memory beyond one second. We have added comments on this aspect in Sec V G of Supplementary Information (lines 360-362).

(7) Concerning the question and suggestion:

“c. Will optically induced NV ionization affect the nuclear spin coherence in the presence of rf driving?

It will be important to include these details in the supplemental material in order to make the claim that combined optical illumination and rf driving can be useful.”

The optically NV ionization prolongs the nuclear spin coherence time, according to the experiment in ref. 32 [Science 336, 1283 (2012)]. The presence of rf driving does not interfere in the optical illumination process, because of their very different operating frequencies. In this

respect, we want to note that in ref. 32 [Science 336, 1283 (2012)], rf decoupling fields have been already applied during optical illumination to further prolong the coherence time of the nuclear memory without any undesired effect. As suggested by the Reviewer, we have included these details in Sec V G of Supplementary Information (lines 362-364).

(8) Regarding the reviewer's question

"3. In Fig.2, is there a particular reason why panel (a) and panel (b) choose different interaction times 10 us and 13 us?"

We thank the Reviewer for the very careful reading. The reason of the queried choices is minor. We selected that time in panel (b) because the differences between the signal of C2 (which is coming from two spins) and the dips coming from the action of one spin were more evident when presented in that 2D plot. To eliminate any potential confusion, in the resubmitted version, we adopt the same interaction time of 13 us in both panels.

(9) Concerning the remark

"Overall, I think that the paper presents interesting ideas that may further extend our capability of individually control more nuclear spins around a single NV center. If the authors can address the above comments and questions, I would like to recommend the publication of this paper in Nature Communications."

We thank Reviewer #1 again for her/his constructive comments and questions that, we believe, have been adequately addressed in this resubmitted version. Therefore, we hope our paper is now ready for publication in Nature Communications.

Response to Reviewer #2

We thank Reviewer #2 for reviewing our manuscript. In the following we respond to all the remarks pointed out by Reviewer #2.

(1) Concerning the reviewer's remark

"The paper of Wang et al., proposes a method for the detection of nuclear spins which are weakly coupled to an electron spin - specifically that of a single NV center in diamond. The idea is based on using a long (and thus highly frequency-selective) RF pulse to drive the nuclear spins, applied either while an electron spin coherence is protected by dynamical decoupling, or while it is being stored in a nuclear spin memory. A find two significant flaws in the current work which prohibit me to recommend publication. "

We believe that the Reviewer does not appreciate the full scope of our work. Specifically, we would like to point out that our method is not only limited to spin detection. More importantly, our

method has the capability to *individually control* many nuclear spins around a single NV center, which is a fundamental requirement for the development of a solid-state quantum processor. We want to note that this is one of the advantages that our method present and it has been confirmed by the other two Reviewers. Furthermore, we do not agree with the two concerns prohibiting Reviewer #2 to recommend publication. In the following lines we demonstrate that these concerns do not stand.

(2) Concerning the remark

“ 1) This proposal falls within a methodology known as "Electron-nuclear double resonance", or ENDOR, in the field of magnetic resonance. A good reference for this is Chapter 12 of "Principles of Pulse EPR" by Schweiger and Jeschke. The manuscript surprisingly does not make any reference to established ENDOR methods, including those which have been designed to detect nuclear spins which are weakly coupled to the electron spin. For example, at first inspection, I would expect the Mims ENDOR sequence (12.2.2 in the book cited above) to yield a similar result to that shown in Fig 2A. Furthermore, the proposed sequence bears a striking resemblance to "Hyperfine-decoupled ENDOR" (12.3.2), albeit using DD rather than spin-locking. I recommend that the authors perform a thorough review of established ENDOR methods (which go back 50 years), and then benchmark their proposal against these methods to determine what tangible advantages their approach might offer. As far as I'm aware, the element of their proposal which uses a nuclear spin memory may be novel, but the range of circumstances where that approach may be practically useful could be limited. “

We thank the reviewer for regarding our proposal with the use of a nuclear memory as novel. We would like to note that long coherence times and good control on nuclear memories have already been demonstrated [e.g., refs. Science 336, 1283 (2012), Nature Nanotech. 9, 171 (2014), Nature 506, 204 (2014), Phys. Rev. X 6, 021040 (2016)], which confirms the potential of using nuclear spin as a reliable quantum memory.

On the other hand, we want to comment that these ENDOR sequences pointed out by Reviewer #2 are quite different from our method. In this respect, it is clearly stated in the reference cited by the Reviewer that both ENDOR sequences introduce nuclear modulation on the electron spin regardless of the rf frequency/detuning, see for example Eqs. (12.2.7) and (12.3.9) in the reference cited by the reviewer (or see the new Sec IV in the revised supplemental information). Therefore, the electron spin is generally coupled to *all* nuclear spins by using these ENDOR sequences, prohibiting the capability of individual nuclear spin control. In contrast, our method has the ability to exert individual control on nuclear spins, and therefore off-resonant nuclear spins do not contribute the signal [see equations (2) and (3) in the resubmitted version]. Note that having individual control is a central requirement for developing a solid-state quantum information processor. Our argument is also supported by Reviewer #3, who agrees that our ideas are novel and fulfill the originality requirement, comparing with established methods in DEER and ENDOR.

As a response to the Reviewer's comments, in the resubmitted version we have included more references (refs. 3,17,18,26,27 in the main text and refs. 30,31 in supplementary information) on ENDOR, including the good book suggested the Reviewer. In addition, to assist the readers and to clarify the differences between our method and the ENDOR sequences, we have added a new section (Sec. IV. Discussion on electron nuclear double resonance, lines 239-282) in Supplementary Information.

(3) Concerning the remark

"2) My second concern is that it would be typical in magnetic resonance, when a pulse sequence is proposed, to accompany that proposal with an experimental demonstration. Sometimes sequences which appear to work well on paper / in silico do not always perform as well in practical due to technical considerations. Therefore, it would be important, in my view, for a work such as this to include some experimental demonstration of the proposed technique, in order for it to attract a wide interest"

While we agree that physics is ultimately an experimental science in that all theories need to stand up to experimental scrutiny and that the usefulness of experimental techniques is ultimately decided by experiment we disagree profoundly that only experimental work should be published in journals such as Nature Communications.

Of course a theoretical proposal needs to take into account as many practical considerations of current experiments as possible, but leave a sufficient degree of generality to allow any experimentalists to adapt the proposed ideas to their setup. Indeed, in our work we have made exactly such an effort to provide convincing evidence that our scheme should be feasible with current experimental technology. Our study unveils that our method possesses an intrinsic robustness that makes it insensitive to rf field fluctuations and initial memory polarizations in sensing. Many practical issues on NV centres are also considered, for example, the perturbations from other strongly nuclear spins, e.g., from the intrinsic nitrogen spin. To further convince the Reviewer #2, in the resubmitted version we have demonstrated by detailed simulations that our proposal still performs very well even in the presence of typical technical errors.

We are therefore confident that our scheme is not only realizable with today's technology but also offers considerable advantages over previous schemes, even in the presence of unavoidable experimental imperfections. Experiment, stimulated by our work, will now need to study our protocol in practical devices. Therefore, we believe that our work merits publication in Nature Communications as a pure theory paper which would allow it reach and stimulate the largest possible set of experimental teams.

Response to Reviewer #3

We thank Reviewer #3 for her/his careful reading and very constructive and insightful comments. In the following we address all her/his comments and questions.

(1) Concerning

“A. The manuscript describes a scheme enhancing the capability of addressing and controlling specific nuclear spins in solids, through their interaction with a probe electronic spin. The method is described in the context of an NV center in diamond coupled to other nuclear spins, mostly ^{13}C spins in the diamond lattice.

While such addressing and control has been demonstrated in the past (as mentioned, e.g., in refs. 3,4,22), the proposed scheme incorporates an additional RF driving of the nuclear spins to provide another control element for enhanced addressability and control.

The authors demonstrate theoretically the advantages of the proposed scheme, along with several improvements, such as combining with dynamical decoupling (DD) of the NV spin and using nearby nuclear spins as memory for better spectral resolution.

B. The proposed scheme could potentially improve significantly the ability to address and control numerous nuclear spins through the control of a single electronic spin, thus impacting applications ranging from quantum information processing to nuclear magnetic resonance. It is therefore of interest to a relatively large community.

While the precise features of the scheme are novel, it is strongly based on known approaches, essentially combining driving of the control spin (the electronic spin in this case) and the target spins (the nuclear spins in this case). Such approaches include DEER and ENDOR schemes from the field of NMR, which have been demonstrated with NV centers as well (e.g., PRL 113, 197601 (2014) and Science 339, 557 (2013)). Nevertheless, I believe that the combined techniques presented in the current paper fulfill the originality requirement.

C., D., E., G. The proposal is clearly presented, along with the numerical analysis and results (with a few caveats mentioned below). Except for the few issues mentioned below, the approach is sound, well written, and convincing. The references are rather comprehensive.”

we thank the Reviewer for the good summary of our work. To further improve the presentation, we have added the references pointed out by the Reviewer, 17 and 18 [PRL 113, 197601 (2014); Science 339, 557 (2013)], in the revised manuscript.

(2) Concerning:

“A few issues: - The authors include DD pulses to enhance the coherence properties of the NV, as described e.g. on line 116. This is a common approach used extensively in the NV community and beyond. For nuclear spin sensing it is common to apply symmetrized sequences, such as the XY family, instead of the CP sequence, which usually results in spurious artifacts in the signal. Why do the authors choose to simulate the CP sequence? They should consider pulse errors, which are usually a major concern in experimental realizations.”

We thank the Reviewer for the constructive remarks. We adopted the CP sequence because many robust periodic sequences, including the XY family, are derived from it and have the same pulse timing. Now, we have commented on the potential use of other sequences (see lines 134-136, 347-348 in the resubmitted version and new references 28,29) and demonstrated the performance of our scheme at the presence of pulse errors, by using equally-spaced adaptive XY-8 (AXY-8) sequences as an example (see the newly added Fig. 3 and related lines 213-249 in the main text and the new Sec. V. C., lines 310-326, in Supplementary Information).

(3) Following the suggestion

“- In standard NV-based nuclear spin measurement schemes, direct RF driving of the nuclear spins is not used, but a signal can be extracted from the combined Larmor precession of the nuclei and their hyperfine interaction with the NV. The authors should clearly compare that approach with the proposed driven scheme, and present the signals on the same footing, e.g. in Fig. 2a.”

We have added the signals of standard DD (upper panel of Fig. 2a) for comparison, along with a discussion in lines 161-179 and 184-189.

(4) Concerning the remark

“- Fig. 2 is rather confusing and poorly explained. It is not clear to me what the different dashed lines in 2a present, and the triangular arrows in the C1-C3 labeling. How many nuclear spins were included in this simulation?”

We apologise for the confusion. The dashed lines indicate the expected height of the signal dip if there are one or two nuclear spins with the precession frequency equaling to the rf frequency. Accordingly, we have revised the caption as well as the lines 145-154 in the main text to explain the dashed lines. In the previous version, the triangular arrows indicate the nuclear precession frequencies and the arrow lengths are proportional to the values of the perpendicular hyperfine components A_j^\perp . Now we have changed the arrows to vertical lines for a better understanding.

Furthermore, in the main text of revised manuscript (line 132), we have commented that there are 736 carbon-13 spins used in the simulation (in addition to the intrinsic nitrogen-14, see lines 301-302 in Supplementary Information).

(5) Concerning the question

“In the caption it is mentioned that the parameters used in Fig. 2a are different than the ones in 2b. Why? It would have been better to use the same parameters. ”

We thank the Reviewer for the very careful reading. The reason of the queried choices is minor. We selected that time in panel (b) because the differences between the signal of C2 (which is coming from two spins) and the dips coming from the action of one spin were more evident

when presented in that 2D plot. To eliminate any potential confusion, in the resubmitted version, we adopt the same interaction time of 13 μ s in both panels.

(6) We thank the reviewer for the insightful observation:

“Also, it's not clear to me why in Fig. 2b there's a distinct difference between 0 phase and 2π phase (in the vertical axis). The authors should explain this difference.”

The fringes are caused by the rectangular rf pulse applied at the delay window. Because the rf pulse has a finite duration, it has side frequencies around its centre carry frequency (approximately, relating to the Fourier transform of an oscillating field with a finite duration). For a given pulse length, a stronger rf field strength causes stronger side fringes, making the distinct difference between a rotation angle of 0 (i.e., vanishing rf field) and a rotation angle of 2π .

In the resubmitted version, we have added a discussion on this aspect (lines 194-212 and 465-469).

(7) Concerning the comment:

“- In Fig. 3a the upper and lower panels describe different memory nuclear spins (^{13}C vs. N), but the results seem almost indistinguishable. This is surprising, and if not a mistake, should be explained in more detail.”

This is not a mistake. Their similarity is because that both their hyperfine interactions are stronger than the inverse of electron spin relaxation time. Now in the revised manuscript we have commented on this aspect (lines 275-282) and explained in more details in Supplemental Information (Sec. III C, lines 219-238).

(8) We thank the reviewer for her/his constructive suggestions:

“- I felt that the details in the main text are lacking, with more detail present in the supplementary. The authors and editors should consider moving some detail from the supplementary to the main text.”

Now we have explained in more details and have moved some details from the supplementary information to the main text (see Summary of Changes).

(9) Concerning the remark:

“- In the supplementary, line 50, it is stated that for ± 1 states the hyperfine shift of the Larmor frequencies vanishes. This is not trivial to me, and should be explained in detail, as it is an important aspect of the theory.”

we appreciate the reviewer for considering this aspect as important. A simplified explanation is that if one performs DD on the $+1$ and -1 states, the average of the hyperfine fields on the

nuclear spins is zero. In the revised version of Supplementary Information (Sections I, lines 56-77 and II B, lines 112-131), we have explained this aspect in detail.

(10) Regarding the summary:

“In summary, I believe that the paper presents an interesting and potentially important scheme for enhanced addressing and control of nuclear spins using an control electronic spin.

I would recommend publication of the manuscript if the comments above are fully resolved.”

We thank the Reviewer for her/his constructive comments and questions, all of which we believe we have adequately addressed. Therefore we hope that she/he is now convinced to recommend the publication of this paper in Nature Communications.

Reviewer #1 (Remarks to the Author):

The authors have convincingly addressed my comments and questions in the revised version of the manuscript.

In addition, I agree with the authors that the proposed method is novel and different from the standard ENDOR sequences.

Therefore, I would like recommend publication of this work in Nature Communications.

Reviewer #2 (Remarks to the Author):

The revised manuscript has now introduced extensive reference to the well established family of ENDOR methods, including a detailed comparison with Mims and PEANUT ENDOR in the supporting information. I am satisfied that the ENDOR pulse sequence proposed in this work is novel, to my knowledge, and offers some potential advantages over established methods, particularly in the entanglement of electron spins and weakly-coupled nuclear spins.

On the other hand, I find the attempts to distinguish this proposal from previous work a bit ham-fisted. First, what is being proposed isn't an alternative to the ENDOR family of techniques, but rather a new ENDOR sequence - indeed one which could be useful in different areas, and not just NV centers. Second, there are several ENDOR methods which already enable the spectroscopic resolution of weakly coupled spins, and although this is reflected by the authors in their detailed response, this point doesn't always come across in the manuscript. For example, in second line of the abstract: "However, with standard techniques, no more than 8 nuclear spins have been resolved by a single defect center. Here we develop a method that improves significantly the ability to spectrally resolve nuclear spins..." In fact, as the authors explain in their letter, existing ENDOR techniques are already capable of resolving weakly coupled spins with the same resolution as in the method presented here. The advantage, as argued by the authors, is in the control and not spectral resolution, and if this is the case it the abstract and introduction of the paper should be written accordingly.

Third, the claim that existing ENDOR/NMR methods are unable "to control individual nuclear spins is confusing" - the ability to control the nuclear spin surely comes from the selectivity of the (long) RF pulse, and there are many ENDOR methods which permit this. I believe what the authors are particularly interested in is a specific form of control capable of entangling the electron spin with a particular weakly-coupled nuclear spin. Again, this qualification is important: following established methods (e.g. the 50-year old Mims ENDOR sequence), weakly-coupled nuclear spins can already be prepared, controlled using arbitrary RF pulses, and measured.

In short, I would argue that this work is really about introducing a new ENDOR method which can be used to apply highly selective entangling gates between an electron and weakly-coupled nuclear spins. The other arguments, more prominent in the current draft, about "spectral resolution" and "individual control" do not really do justice to well-established methods.

So, if this paper were presented in this light, as a new method for achieving highly selective entangling gates, would it merit publication in Nature Communications? Certainly the work is extremely thorough, and very well presented with excellent figures and I commend the authors for a very polished piece of work. Although I fully agree with the authors that purely theoretical works are absolutely appropriate in Nature Communications, I would expect such papers to include a degree of novel science or outline a powerful new technique which could stimulate an ambitious experimental programme. The fact is that the paper does not contain much (any?) new physics, and as a proposal for a new practical method, it would be trivial for any one of the dozens of experimental groups working on control of NV-center spins to implement. For this reason, I cannot see the justification for not simply implementing the scheme to see if it works as well as the authors claim, and do not believe this particular theoretical proposal on its own justifies publication

at the broad level of Nature Communications.

Reviewer #3 (Remarks to the Author):

The authors have addressed all the issues that I raised in my review, as well as the comments of the other referees.

It is my opinion that as a result that manuscript is now improved, well-written and clearly conveys the novel and appealing outcomes of the research.

I therefore recommend that manuscript for publication in Nature Communications.

Summary of changes:

We include a pdf version where all changes are highlighted in blue. We have fixed the styles to the requirements of Nature Communications, such as the 150 word-count limit for the abstract and labels for the supplementary material. Two colored lines in Figure 2a have been changed to avoid confusion for color-blind readers (now the colors of the lines become magenta and turquoise). We have revised the abstract, introduction, and discussion parts to highlight the difference between our method and existing techniques for manipulating nuclear spins.

Reply to Reviewer #1:

Report:

The authors have convincingly addressed my comments and questions in the revised version of the manuscript. In addition, I agree with the authors that the proposed method is novel and different from the standard ENDOR sequences. Therefore, I would like recommend publication of this work in Nature Communications.

Response:

We thank the Reviewer for recommending publication of our work in Nature Communications, as well as for the highlight that our method “is novel and different from the standard ENDOR sequences”.

Reply to Reviewer #3:

Report:

The authors have addressed all the issues that I raised in my review, as well as the comments of the other referees. It is my opinion that as a result that manuscript is now improved, well-written and clearly conveys the novel and appealing outcomes of the research. I therefore recommend that manuscript for publication in Nature Communications.

Response:

We thank the Reviewer for recommending publication of our work in Nature Communications, as well as for the remark that the manuscript “is now improved, well-written and clearly conveys the novel and appealing outcomes of the research”.

Reply to Reviewer #2:

(1) Report:

The revised manuscript has now introduced extensive reference to the well established family of ENDOR methods, including a detailed comparison with Mims and PEANUT ENDOR in the supporting information. I am satisfied that the ENDOR pulse sequence proposed in this work is

novel, to my knowledge, and offers some potential advantages over established methods, particularly in the entanglement of electron spins and weakly-coupled nuclear spins.

Response:

We thank the Reviewer for reviewing our manuscript again, as well as for saying that our method offers advantages over established methods, particularly in the control of weakly-coupled nuclear spins by the centre electron.

(2) Report:

On the other hand, I find the attempts to distinguish this proposal from previous work a bit ham-fisted.

First, what is being proposed isn't an alternative to the ENDOR family of techniques, but rather a new ENDOR sequence - indeed one which could be useful in different areas, and not just NV centers.

Second, there are several ENDOR methods which already enable the spectroscopic resolution of weakly coupled spins, and although this is reflected by the authors in their detailed response, this point doesn't always come across in the manuscript. For example, in second line of the abstract: "However, with standard techniques, no more than 8 nuclear spins have been resolved by a single defect center. Here we develop a method that improves significantly the ability to spectrally resolve nuclear spins..." In fact, as the authors explain in their letter, existing ENDOR techniques are already capable of resolving weakly coupled spins with the same resolution as in the method presented here. The advantage, as argued by the authors, is in the control and not spectral resolution, and if this is the case it the abstract and introduction of the paper should be written accordingly.

Third, the claim that existing ENDOR/NMR methods are unable "to control individual nuclear spins is confusing" - the ability to control the nuclear spin surely comes from the selectivity of the (long) RF pulse, and there are many ENDOR methods which permit this. I believe what the authors are particularly interested in is a specific form of control capable of entangling the electron spin with a particular weakly-coupled nuclear spin. Again, this qualification is important: following established methods (e.g. the 50-year old Mims ENDOR sequence), weakly-coupled nuclear spins can already be prepared, controlled using arbitrary RF pulses, and measured.

In short, I would argue that this work is really about introducing a new ENDOR method which can be used to apply highly selective entangling gates between an electron and weakly-coupled nuclear spins. The other arguments, more prominent in the current draft, about "spectral resolution" and "individual control" do not really do justice to well-established methods.

Response:

We thank the Reviewer for the suggestions on distinguishing our method from existing ENDOR techniques. We also thank her/him for regarding our method with wide applications “in different areas, and not just NV centers”.

We would like to highlight that “demonstrating spectroscopy resolvability does not necessarily imply controllability”, as it was stated by Reviewer #1 in the first remark of her/his first report. If a method does not eliminate the noise from unwanted spins (e.g., those not precessing at the rf control frequency), the signal from target spins can be strongly perturbed by the noise. As we already demonstrated in Supplementary Note 4, *even one nuclear spin* already significantly perturbs the signal, because the Mims ENDOR is not designed to eliminate spin noise. The bath spin noise also unavoidably reduces the coherence times of sensor electron and reduces the sensitivity. It is for this reason that detection and control of nuclear spins can be limited to nuclear spins with distinct characters and with pronounced interactions compared with other bath spins. This argument is supported by the very-recent attempts to apply ENDOR techniques in NV centres (where the NV electron could be coupled to dozens of nuclear spins). In those experiments, e.g., Refs. 23, 26, 27, the spectral resolution is improved for spins with the strongest coupling but there was no clear demonstration on the increase in the detectable nuclear spins compared with the results by DD techniques. In summary, known ENDOR methods only have the ability to detect a small number of spins, but our method can efficiently detect, and address and manipulate a large number of nuclear spins individually.

Following the Reviewer's suggestions, we have revised the abstract, introduction, and the discussion section accordingly to clarify existing techniques and their difference between our method.

(3) Report:

So, if this paper were presented in this light, as a new method for achieving highly selective entangling gates, would it merit publication in Nature Communications? Certainly the work is extremely thorough, and very well presented with excellent figures and I commend the authors for a very polished piece of work. Although I fully agree with the authors that purely theoretical works are absolutely appropriate in Nature Communications, I would expect such papers to include a degree of novel science or outline a powerful new technique which could stimulate an ambitious experimental programme. The fact is that the paper does not contain much (any?) new physics, and as a proposal for a new practical method, it would be trivial for any one of the dozens of experimental groups working on control of NV-center spins to implement. For this reason, I cannot see the justification for not simply implementing the scheme to see if it works as well as the authors claim, and do not believe this particular theoretical proposal on its own justifies publication at the broad level of Nature Communications.

Response:

We thank the Reviewer for regarding our method “for achieving highly selective entangling gates” and our work as “extremely thorough, and very well presented with excellent figures”.

Our work reveals an important piece of physical point overlooked in the ENDOR methods: selectively removing the contribution from spins with frequencies different from the rf frequency is a key to increase the number of detectable/controllable spins. Given that our theoretical protocol is easy to implement (e.g., considering that it requires only standard techniques as the 50-year old ENDOR sequences) as well as its unique abilities we have discussed, our method is important to many fields including quantum technologies, even without experimental demonstration. Regarding its potential impacts “in different areas, and not just NV centers” stated by Reviewer #2, our work should justify publication at Nature Communications, which is also supported by other two Reviewers.